# GShard: Scaling Giant Models with Conditional Computation and Automatic Sharding

**Dmitry Lepikhin**
lepikhin@google.com

**HyoukJoong Lee**
hyouklee@google.com

**Yuanzhong Xu**
yuanzx@google.com

**Dehao Chen**
dehao@google.com

**Orhan Firat**
orhanf@google.com

**Yanping Huang**
huangyp@google.com

**Maxim Krikun**
krikun@google.com

**Noam Shazeer**
noam@google.com

**Zhifeng Chen**
zhifengc@google.com

## Abstract

Neural network scaling has been critical for improving the model quality in many real-world machine learning applications with vast amounts of training data and compute. Although this trend of scaling is affirmed to be a sure-fire approach for better model quality, there are challenges on the path such as the computation cost, ease of programming, and efficient implementation on parallel devices. In this paper we demonstrate conditional computation as a remedy to the above mentioned impediments, and demonstrate its efficacy and utility. We make extensive use of GShard, a module composed of a set of lightweight annotation APIs and an extension to the XLA compiler to enable large scale models with up to trillions of parameters. GShard and conditional computation enable us to scale up multilingual neural machine translation Transformer model with Sparsely-Gated Mixture-of-Experts. We demonstrate that such a giant model with 600 billion parameters can efficiently be trained on 2048 TPU v3 cores in 4 days to achieve far superior quality for translation from 100 languages to English compared to the prior art.

## 1 Introduction

Scaling neural networks brings dramatic quality gains over a wide array of machine learning problems such as computer vision, language understanding and neural machine translation (Devlin et al., 2018; Mahajan et al., 2018; Arivazhagan et al., 2019; Huang et al., 2019; Brown et al., 2020b). This general tendency motivated recent studies to scrutinize the factors playing a critical role in the success of scaling, including the amounts of training data, the model size, and the computation being utilized as found by past studies (Advani & Saxe, 2017; Hestness et al., 2019; Geiger et al., 2020). While the final model quality was found to have a power-law relationship with these factors (Hestness et al., 2017; Kaplan et al., 2020), the significant quality gains brought by larger models also came with various practical challenges. *Training efficiency*, which we define as the amount of compute and time used to achieve a superior model quality against the best system existed, is oftentimes left out.

In this study, we strive for improving the model quality while being training efficiently. We built a 600 billion parameters sequence-to-sequence Transformer model with Sparsely-Gated Mixture-of-Experts layers, which enjoys sub-linear computation cost and $O(1)$ compilation time. We trained this model with 2048 TPU v3 devices for 4 days on a multilingual machine translation task and achieved far superior translation quality compared to prior art when translating 100 languages to English with a single non-ensemble model. We conducted experiments with various model sizes and found that the translation quality increases as the model gets bigger, yet the total wall-time to train only increases sub-linearly with respect to the model size, as illustrated in Figure 1. To train such an extremely large model, we relied on the following key design choices.

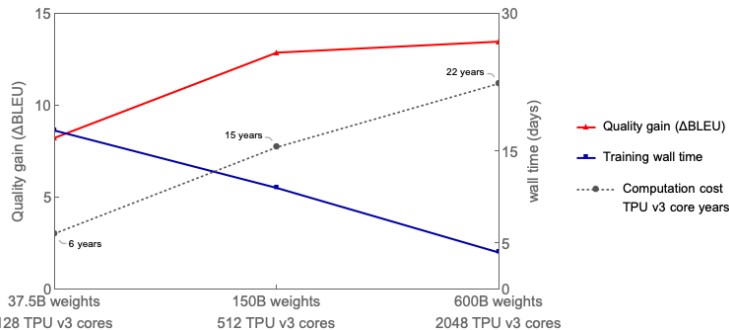

**Figure 1:** Multilingual translation quality (average $\Delta$BLEU comparing to bilingual baselines) improved as MoE model size grows up to 600B, while the end-to-end training cost (in terms of TPU v3 core-year) only increased sublinearly. Increasing the model size from 37.5B to 600B (16x), results in computation cost increase from 6 to 22 years (3.6x). The 600B parameters model that achieved the best translation quality was trained with 2048 TPU v3 cores for 4 days, a total cost of 22 TPU v3 core-years. In contrast, training all 100 bilingual baseline models would have required 29 TPU v3 core-years. Our best quality dense single Transformer model (2.3B parameters) achieving $\Delta$BLEU of 6.1, was trained with GPipe for a total of 235.5 TPU v3 core-years.

**Conditional computation** First, model architecture should be designed to keep the computation and communication requirements sublinear in the model capacity. Conditional computation enables us to satisfy training and inference efficiency by having a sub-network activated on the per-input basis. Shazeer et al. (2017) has shown that scaling RNN model capacity by adding Sparsely Gated Mixture-of-Experts (MoE) layers allowed to achieve improved results with sub-linear cost. We therefore present our approach to extend Transformer architecture with MoE layers in this study.

**GShard Annotation** Second, the model description should be separated from the partitioning implementation and optimization. This separation of concerns let model developers focus on the network architecture and flexibly change the partitioning strategy, while the underlying system applies semantic-preserving transformations and implements efficient parallel execution. To this end we propose a module, GShard, which only requires the user to annotate a few critical tensors in the model with partitioning policies. It consists of a set of simple APIs for annotations, and a compiler extension in XLA for automatic parallelization. Model developers write models as if there is a single device with huge memory and computation capacity, and the compiler automatically partitions the computation for the target based on the user annotations and their own heuristics.

## 2 MODEL

The Transformer (Vaswani et al., 2017) architecture has been widely used for natural language processing. We scale Transformer with conditional computation by replacing every other feed-forward layer with a sparsely activated Position-wise Mixture of Experts (MoE) layer (Shazeer et al., 2017), with a variant of top-2 gating in both the encoder and the decoder (Figure 2). Each subword token in the training example activates a sub-network of the MoE Transformer during both training and inference. The size of the sub-network is roughly independent of the number of experts per MoE Layer, allowing sublinear scaling of the computation cost.

### 2.1 POSITION-WISE MIXTURE-OF-EXPERTS LAYER

The Mixture-of-Experts (MoE) layers used in our model differ from Shazeer et al. (2017)'s in the sparse gating function and the auxiliary loss being used. A MoE layer for Transformer consists of $E$ feed-forward networks $\text{FFN}_1 \ldots \text{FFN}_E$, each of which outputs $wo_e \cdot \text{ReLU}(wi_e \cdot x_s)$, where $x_s$ is the input token to the MoE layer, $wi$ and $wo$ being the input and output projection matrices for the feed-forward layer (an expert) with shapes $[M, H]$ and $[H, M]$, respectively. The output of a MoE layer is the combination of the expert outputs $\sum_{e=1}^{E} \mathcal{G}_{s,e} \cdot \text{FFN}_e(x_s)$, where the vector $\mathcal{G}_{s,E}$ is computed by a gating function $\text{GATE}(\cdot)$. We choose to let each token dispatched to at most two experts. The corresponding gating entries $\mathcal{G}_{s,e}$ become non-zeros, representing how much an expert contributes to the final network output.

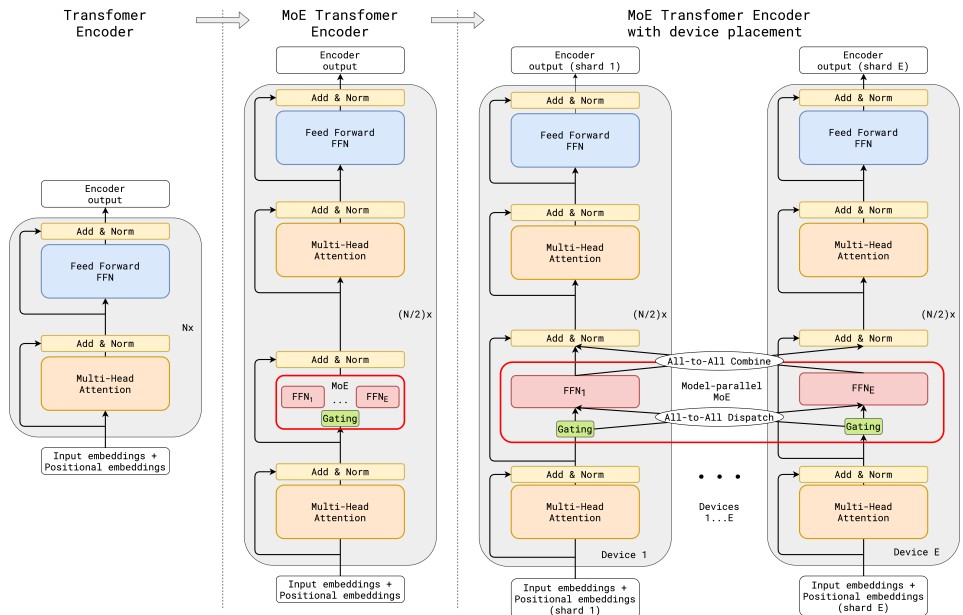

**Figure 2:** Illustration of scaling of MoE Transformer Encoder Layers. Decoder modification is similar. (a) Standard Transformer. (b) Replacing every other feed forward layer with a MoE layer (c) The MoE layer is sharded across multiple devices, while all other layers are replicated.

The gating function GATE($\cdot$) is critical to the MoE layer, which is modeled by a softmax activation function to indicate the weights of each expert in processing incoming tokens. We designed a novel efficient gating function with the following mechanisms (details illustrated in Algorithm 1).

**Load balancing** Naively picking top-$k$ experts from the softmax probability distribution leads to load imbalance problem for training as shown in Shazeer et al. (2017). Most tokens would have been dispatched to a small number of experts, leaving other experts insufficiently trained. To ensure the load is balanced, we enforce that the number of tokens processed by one expert is below some uniform threshold called expert capacity. Assuming $N$ total tokens in a batch and at most two experts per token, then the expert capacity $C$ is set to be $O(N/E)$. GATE($\cdot$) keeps a running counter $c_e$ for how many tokens are dispatched to an expert. When both experts selected by a token already exceed their capacity, the token is considered as an *overflowed* token, where $\mathcal{G}_{s,E}$ degenerates into a zero vector. Such tokens will be passed on to the next layer via residual connections. The introduction of the fixed expert capacity instead of loading balancing functions in Shazeer et al. (2017) allows us to run parallel execution of gating function as described blow.

**Local dispatching for parallel gating** Load balancing required the token assignments of one expert dependent on assignments of the other experts. The original gating function proposed by (Shazeer et al., 2017) had to be implemented sequentially, especially under the static shape constraints on TPUs. In our study, we distributed thousands of experts over thousands of devices, a sequential implementation of the gating function would keep most of the devices idle most of the time. Instead, we propose a new GATE($\cdot$) function that partitions all tokens in a training batch evenly into $G$ local groups, i.e., each group contains $S = N/G$ tokens for local dispatching. All local groups are processed independently in parallel. Each group is given a fractional capacity of each expert, $C = 2N/(G \cdot E)$, to ensure that at most this many tokens are dispatched to an expert. In general, increasing the expect capacity $C$ decreases the number of overflowed tokens thus improves the model quality. Since $G \times C$ is a constant, however, the higher capacity leads to smaller number of groups which hurts the training throughput by limiting the number of parallel gating execution. In this way, we can ensure that expert capacity is still enforced and the overall load is balanced. With fixed expert capacity and local dispatching, we are able to speed up the gating function by $O(G)$ times.

**Auxiliary loss** Following Shazeer et al. (2017), we define a new differentiable auxiliary loss term $\ell_{aux}$ to enforce the load balancing. It is added to the overall loss function of the model $\mathcal{L} = \ell_{ori} + k * \ell_{aux}$ with a constant multiplier $k$, where $\ell_{aux}$ is defined in line (13) of algorithm 1, and the term $c_e/S$ represents the fraction of input routed to each expert. We replace the mean square $(c_e/S)^2$ with

---

**Algorithm 1:** Group-level top-2 gating with auxiliary loss

**Data:** $x_S$, a group of tokens of size $S$
**Data:** $C$, Expert capacity allocated to this group
**Result:** $\mathcal{G}_{S,E}$, group combine weights
**Result:** $\ell_{aux}$, group auxiliary loss

(1) **for** $e \leftarrow 1$ **to** $E$ **do**
(2)   $\quad c_e \leftarrow 0$            ▷ gating decisions per expert
(3)   $\quad g_{S,e} \leftarrow softmax(wg \cdot x_S)$     ▷ gates per token per expert, $wg$ are trainable weights
(4)   $\quad m_e \leftarrow \frac{1}{S} \sum_{s=1}^{S} g_{s,e}$         ▷ mean gates per expert
(5) **end**
(6) **for** $s \leftarrow 1$ **to** $S$ **do**
(7)   $\quad g1, e1, g2, e2 = top\_2(\{g_{s,e} | e = 1 \cdots E\})$     ▷ top-2 gates and expert indices
(8)   $\quad g1 \leftarrow g1/(g1 + g2)$         ▷ normalized $g1$
(9)   $\quad c \leftarrow c_{e1}$         ▷ position in $e1$ expert buffer
(10)   $\quad$ **if** $c_{e1} < C$ **then**
(11)   $\quad\quad \mathcal{G}_{s,e1} \leftarrow g1$         ▷ $e1$ expert combine weight for $x_s$
(12)   $\quad$ **end**
(13)   $\quad c_{e1} \leftarrow c + 1$         ▷ incrementing $e1$ expert decisions count
(14) **end**
(15) $\ell_{aux} = \frac{1}{E} \sum_{e=1}^{E} \frac{c_e}{S} \cdot m_e$
(16) **for** $s \leftarrow 1$ **to** $S$ **do**
(17)   $\quad g1, e1, g2, e2 = top\_2(\{g_{s,e} | e = 1 \cdots E\})$     ▷ top-2 gates and expert indices
(18)   $\quad g2 \leftarrow g2/(g1 + g2)$         ▷ normalized $g2$
(19)   $\quad rnd \leftarrow uniform(0, 1)$     ▷ dispatch to second-best expert with probability $\propto 2 \cdot g2$
(20)   $\quad c \leftarrow c_{e2}$         ▷ position in $e2$ expert buffer
(21)   $\quad$ **if** $c < C \wedge 2 \cdot g2 > rnd$ **then**
(22)   $\quad\quad \mathcal{G}_{s,e2} \leftarrow g2$         ▷ $e2$ expert combine weight for $x_s$
(23)   $\quad$ **end**
(24)   $\quad c_{e2} \leftarrow c + 1$
(25) **end**

---

differentiable approximation $m_e(c_e/S)$, which can provide better numerical stability since it can be optimized with gradient descent.

**Random routing** Intuitively, the output $y_s$ is a weighted average of what selected experts return. If the weight for the 2nd expert is very small, we can simply ignore the 2nd expert to conserve the overall expert capacity. Hence, in addition to respecting the expert capacity constraint, GATE($\cdot$) dispatches to the 2nd-best expert with the probability proportional to its weight $g_2$. We observed much less overflowed tokens thus better accuracy with random routing for models at the small scale. We then adopted this approach for our experiments at large scales.

## 2.2 HIGHLY PARALLEL IMPLEMENTATION USING GSHARD

To implement the model in Section 2.1 efficiently on a cluster of devices, we first express the model in terms of linear algebra operations, which are highly tailored and optimized in our software stack TensorFlow (Abadi et al., 2016) and the hardware platform (TPU).

Our model implementation (Algorithm 2) views the whole accelerator cluster as a single device and expresses its core algorithm in a few tensor operations independent of the setup of the cluster. We extensively used tf.einsum, the Einstein summation notation (Einstein, 1923), to concisely express the model. *Top2Gating* in Algorithm 2 computes the union of all group-local $\mathcal{G}_{S,E}$ described in the gating Algorithm 1. *combine_weights* is a 4-D tensor with shape $[G, S, E, C]$, whose element value becomes non-zero when the input token $s$ in group $g$ is sent to expert $e$ at capacity buffer position $c$. For a specific $g$ and $s$, a slice *combine_weight[g, s, :, :]* contains at most two non-zero values. Binary *dispatch_mask* is produced from *combine_weights* by simply setting all non-zero values to 1.

To scale the computation to a cluster with $D$ devices, we choose the number of groups $G$ and the number of experts $E$ proportional to $D$. With $CE = O(2S)$ and the number of tokens per group $S$ independent of $D$, the model dimension $M$ and the feed-forward hidden dimension $H$, the total

number of floating point operations (FLOPS) per device in Algorithm 2:

$$
\begin{aligned}
&FLOPS_{\text{Softmax}} + FLOPS_{\text{Top2Gating}} + FLOPS_{\text{Dispatch|Combine}} + FLOPS_{\text{FFN}} \\
&= O(GSME)/D + O(GSEC)/D \quad + O(GSMEC)/D \quad + O(EGCHM)/D \\
&= O(DM) \qquad\quad + O(2) \qquad\qquad + O(2M) \qquad\qquad + O(2HM)
\end{aligned}
$$

---

**Algorithm 2:** Forward pass of the Positions-wise MoE layer. The underscored letter (e.g., G and E) indicates the dimension along which a tensor will be partitioned.

```
1  gates = softmax(einsum("GSM,ME->GSE", inputs, wg))
2  combine_weights, dispatch_mask = Top2Gating(gates)
3  dispatched_inputs = einsum("GSEC,GSM->EGCM", dispatch_mask, inputs)
4  h = einsum("EGCM,EMH->EGCH", dispatched_inputs, wi)
5  h = relu(h)
6  expert_outputs = einsum("EGCH,EHM->GECM", h, wo)
7  outputs = einsum("GSEC,GECM->GSM", combine_weights, expert_outputs)
```

---

The per device flops for softmax is proportional to $D$, but in our experiments $D \leq 2H$ for up to 16K devices so it is less than that of FFN. Consequently the total per-device $FLOPS$ could be considered independent of $D$, satisfying sublinear scaling design requirements. In addition to the computation cost, dispatching and combining token embedding using AllToAll operators consumed $O(\sqrt{D})$ cross-device communication cost on our 2D TPU cluster. We will discuss the cost analysis and micro-benchmarks for such communication overheads in Appendix section A.3.3.

Due to the daunting size and computation demand of tensors in Algorithm 1 when we scale the number of tokens $N$ to millions and the number of experts $E$ to thousands, we have to parallelize the algorithm over many devices. To express parallelism, tensors in the linear algebra computation are annotated with sharding information using GShard APIs to selectively specify how they should be partitioned across a cluster of devices. For example, the underscored letters in Algorithm 2 specified along which dimension the tensors are partitioned. This sharding information is propagated to the compiler so that the compiler can automatically apply transformations for parallel execution. Please refer to appendix A.2 for more detailed description of the GShard module.

We express the annotated version of Algorithm 2 as below. The input tensor is split along the first dimension and the gating weight tensor is *replicated*. After computing the dispatched expert inputs, we apply *split* to change the sharding from the group ($G$) dimension to the expert ($E$) dimension.

```
1     # Partition inputs along the first (group G) dim across D devices.
2   + inputs = split(inputs, 0, D)
3     # Replicate the gating weights across all devices
4   + wg = replicate(wg)
5     gates = softmax(einsum("GSM,ME->GSE", inputs, wg))
6     combine_weights, dispatch_mask = Top2Gating(gates)
7     dispatched_inputs = einsum("GSEC,GSM->EGCM", dispatch_mask, inputs)
8     # Partition dispatched inputs along expert (E) dim.
9   + dispatched_inputs = split(dispatched_inputs, 0, D)
10    h = einsum("EGCM,EMH->EGCH", dispatched_inputs, wi)
```

where *split(tensor, d, D)* annotates tensor to be partitioned along the *d* dimension over D devices, and *replicate(tensor)* annotates tensor to be replicated across partitions. The invocations of GShard APIs such as *split* or *replicate* only adds sharding information to the tensor and does not change its logical shape. Moreover, users are not required to annotate every tensor in the program. Annotations are typically only required on a few important operators like Einsums in our model and the compiler uses iterative data-flow analysis to infer sharding for the rest of the tensors.

## 3 MASSIVELY MULTILINGUAL, MASSIVE MACHINE TRANSLATION (M4)

We chose multilingual neural machine translation (MT) (Firat et al., 2016; Johnson et al., 2017; Aharoni et al., 2019) to validate our design for efficient training with GShard. Multilingual MT,

which is an inherently multi-task learning problem, aims at building a single neural network for the goal of translating multiple language pairs simultaneously. This extends the line of work Huang et al. (2019); Arivazhagan et al. (2019); Shazeer et al. (2017) towards a universal machine translation model (Bapna & Firat, 2020), a single model that can translate between more than hundred languages.

In this section, we advocate how conditional computation (Bengio et al., 2013; Davis & Arel, 2013) with sparsely gated mixture of experts fits into the above detailed desiderata and show its efficacy by scaling neural machine translation models, while keeping the training time of such massive networks practical. E.g. a 600B GShard model for M4 can process 1T tokens (source side tokens after sub-word segmentation) in 250k training steps under 4 days. We experiment with increasing the model capacity by adding more layers and more experts into the model and study the factors playing role in convergence, model quality and training efficiency. Further, we demonstrate how conditional computation can speed up the training and how sparsely gating each token through the network can efficiently be learned without any prior knowledge on task or language relatedness, exemplifying the capability of learning the gating decision directly from the data.

We focus on improving the translation quality (measured in terms of BLEU score Papineni et al. (2002)) from all 100 languages to English. This resulted in approximately 13 billion training examples to be used for model training. Our baselines are separate bilingual Neural Machine Translation models for each language pair (e.g. a single model for German-to-English), tuned depending on the available training data per-language[1]. Rather than displaying individual BLEU scores for each language pair, we follow the convention of placing the baselines along the $x$-axis at zero, and report the $\Delta$BLEU trendline of each massively multilingual model trained with GShard (see Figure 3). The $x$-axis in Figure 3 is sorted from left-to-right in the decreasing order of amount of available training data, where the left-most side corresponds to high-resourced languages, and low-resourced languages on the right-most side respectively. We also include a variant of dense 96 layer Transformer Encoder-Decoder network T(96L) trained with GPipe pipeline parallelism on the same dataset as another baseline, which took over 6 weeks to convergence on 2048 TPU v3 cores [2].

We varied the depth of the transformer network (L) and the number of experts (E) to scale the model. For depth, we tested three different options, 12 (original Transformer depth, which consists of 6 encoder and 6 decoder layers), 36 and 60 layers. For the number of experts that replaces every other feed-forward layer, we also tested three options, namely 128, 512 and 2048 experts. Note that, the number of devices used for training, is fixed to be equal to the number of experts per-layer for simplicity. Please also see the detailed description in Table 1 for model configurations. During training, we use float32 for both model weights and activations in order to ensure training stability. We also ran additional scalability experiments with MoE(2048E, 60L) with bfloat16 activations with more than one trillion model weights. We are still working on the model convergence and hence did not include the results from this trillion weight model for the sake of reproducibility.

## 3.1 RESULTS

For each experiment (rows of the Table 1), we trained the corresponding MoE Transformer model until it has seen 1 trillion ($10^{12}$) tokens. The model checkpoint at this point is used in the model evaluation. We did not observe any over-fitting patterns by this point in any experiment. Instead, we observed that the training loss continued to improve if we kept training longer. We evaluated BLEU scores that the models achieved for all language pairs on a held-out test set in Figure 3.

Here we discuss the implication of each experiment on languages that have large amounts of training data (high resourced), as well as languages with limited data (low-resource). In order to improve the quality for both high- and low-resource languages simultaneously within a single model, scaled models must mitigate *capacity bottleneck* issue by allocating enough capacity to high-resource tasks, while amplifying the *positive transfer* towards low-resource tasks by facilitating sufficient parameter sharing. We loosely relate the expected learning dynamics of such systems with the long-standing memorization and generalization dilemma, which is recently studied along the lines of width vs depth scaling efforts (Cheng et al., 2016). Not only do we expect our models to generalize better to the

---

[1]We tuned batch-size and different values of regularization methods (e.g. dropout) in a Transformer-Big or Transformer-Base layout, for high or low-resourced languages respectively.

[2]T(96L) measured to be processing 1+ trillion tokens at 300k steps, processing around 4M tokens/step

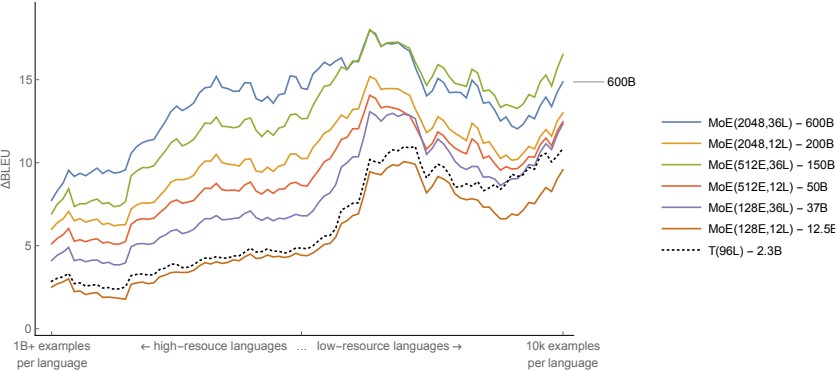

**Figure 3:** Translation quality comparison of multilingual MoE Transformer models trained with GShard and monolingual baselines. MoE(128E, 12L) represents the model with 12 layers and 128 experts per layer. Positions along the $x$-axis represent languages, raging from high- to low-resource. $\Delta$BLEU represents the quality gain of a single multilingual model compared to a monolingual Transformer model trained and tuned for a specific language. MoE Transformer models trained with GShard are reported with solid trend-lines. Dashed trend-line represents a single 96 layer multilingual Transformer model T(96L) trained with GPipe on same dataset. Each trend-line is smoothed by a sliding window of 10 for clarity. (Best seen in color)

held-out test sets, we also expect them to exhibit high transfer capability across languages as another manifestation of generalization performance Lampinen & Ganguli (2018).

**Deeper Models Bring Consistent Quality Gains Across the Board**. We first investigate the relationship between the model depth and the model quality for both high- and low-resource languages. With an increasing number of per-layer experts for each experiment (128, 512 and 2048), we tripled the depth of the network for each expert size, from 12 to 36. Fig. 3 show that when the number of experts per-layer is fixed, increasing the depth (L) alone brings consistent gains for both low and high resourced languages (upwards $\Delta$ shift along the $y$-axis), almost with a constant additive factor every time we scale the depth from 12L to 36L (2-to-3 BLEU points on average in Table 1).

**Relaxing the Capacity Bottleneck Grants Pronounced Quality Gains**. We also consider three models with identical depths (12L), with increasing number of experts per-layer: 128, 512 and 2048. As we increase the number of experts per-layer from 128 to 512, we notice a large jump in model quality, +3.3 average BLEU score across 100 languages. However again by four folds scaling of the number of experts per-layer, from 512 to 2048, yields only +1.3 average BLEU scores. Despite the significant quality improvement, this drop in gains hints the emergence of diminishing returns.

Given the over 100 languages considered, the multilingual model has a clear advantage on improving the low-resource tasks. On the contrary, for high-resource languages the increased number of tasks limits per-task capacity within the model, resulting in lower translation quality compared to a models trained on a single language pair. We observed in our experiments that this *capacity bottleneck* on task interference for high resourced languages can be relaxed by increasing the number of experts per-layer,. Interestingly increasing the depth does not help as much if the capacity bottleneck is not relaxed. For 12 layer models increase in the expert number yields larger gains for high resourced languages as opposed to earlier revealed diminishing returns for low-resourced languages. While adding more experts relaxes the capacity bottleneck, at the same time it reduces the amount of transfer due to a reduction of the shared sub-networks. Notably, $\Delta$BLEU gains for MoE(512E, 36L) exceed ones with higher capacity, but shallower MoE(2048E, 12L). While a comparison of proportionally smaller models, shows that MoE(128E, 36L) is suboptimal compared to MoE(512E, 12L). One can conclude that scaling depth brings most quality gains only after capacity bottleneck is resolved.

**Deep-Dense Models are Better at Positive Transfer towards Low-Resource Tasks.** Lastly we look into the impact of the depth on low-resourced tasks as a loose corollary to our previous experiment. We include a dense model with 96 layers T(96L) trained with GPipe on the same data into our analysis. We compare T(96L) with the shallow MoE(128E, 12L) model. While the gap

| Model | Cores | Steps | Batch sz | TPU core | Training | BLEU | Billion tokens to cross-entropy of | | |
|---|---|---|---|---|---|---|---|---|---|
| | | / sec. | (Tokens) | years | days | avg. | 0.7 | 0.6 | 0.5 |
| MoE(2048E, 36L) | 2048 | 0.72 | 4M | 22.4 | **4.0** | **44.3** | 82 | 175 | 542 |
| MoE(2048E, 12L) | 2048 | 2.15 | 4M | 7.5 | 1.4 | 41.3 | 176 | 484 | 1780 |
| MoE(512E, 36L) | 512 | 1.05 | 1M | 15.5 | 11.0 | 43.7 | 66 | 170 | 567 |
| MoE(512E, 12L) | 512 | 3.28 | 1M | 4.9 | 3.5 | 40.0 | 141 | 486 | - |
| MoE(128E, 36L) | 128 | 0.67 | 1M | 6.1 | 17.3 | 39.0 | 321 | 1074 | - |
| MoE(128E, 12L) | 128 | 2.16 | 1M | 1.9 | 5.4 | 36.7 | 995 | - | - |
| T(96L) | 2048 | - | 4M | ∼235.5 | ∼42 | 36.9 | - | - | - |
| Bilingual Baseline | - | - | | ∼ 29 | - | 30.8 | - | - | - |

**Table 1:** Performance of MoE models with different number of experts and layers.

between the two models measured to be almost constant for the majority of the high-to-mid resourced languages, the gap grows in favor of the dense-deep T(96L) model as we get into the low-resourced regime. Following our previous statement, as the proportion of the shared sub-networks across tasks increase, which is 100% for dense T(96L), the bandwidth for transfer gets maximized and results in a comparably better quality against its shallow counterpart. The same transfer quality to the low-resourced languages can be also achieved with MoE(128E, 36L) which has 37 billion parameters.

We conjecture that, increasing the depth might potentially increase the extent of transfer to low-resource tasks hence generalize better along that axis. But we also want to highlight that the models in comparison have a disproportionate training resource requirements. We again want to promote the importance of *training efficiency*, which is the very topic we studied next.

## 3.2 TRAINING EFFICIENCY

To measure the *training efficiency*. we first keep track of the number of tokens being processed to reach a certain training loss and second we keep track of the wall-clock time for a model to process certain number of tokens. We focus on measuring the training time to fixed training loss targets[3] while varying other factors. We left systems performance analysis in appendex A.3.

**Deeper models converge faster with fewer examples.** It has been shown that, deeper models are better at sample efficiency, reaching better training/test error given the same amount of training examples (Huang et al., 2019; Shoeybi et al., 2019), commonly attributed to the acceleration effect of over-parametrization (Arora et al., 2018). We empirically test the hypothesis again using GShard with MoE Transformers and share trade-offs for models that are not only deep, but also sparsely activated.

For this purpose, we compare number of tokens being processed by each model to reach a preset training loss. A general trend we observe from Table 1 is that, MoE Transformer models with 3 times the depth need 2 to 3 times fewer tokens to reach the preset training loss thresholds. For example MoE(128E, 12L) takes 3 times the number of tokens to reach 0.7 training cross-entropy compared to MoE(128E, 36L). We observe a similar trend for models with 512 and 2048 experts.

Another intriguing observation from Table 1, is again related to the presence of *capacity bottleneck*. Comparing the models with same depth, we notice a significant drop in the number of tokens required to reach training loss of 0.7, as we transition from 128 to 512 number of experts. Practically that is where we observed the capacity bottleneck was residing. After this phase shift, models with ample capacity tend to exhibit similar sample efficiency characteristics.

**Model with 600B parameters trained under 4 days achieved the best quality.** Next we delve deeper into the interaction between model size and wall-clock time spent for training. We monitor number of TPU cores being used, training steps per-second, total number of tokens per batch, TPU core years[4], and actual wall-clock time spent in days for training (see Table 1 columns respectively). One of the largest models we trained, MoE(2048E, 36L) with 600 billion parameters, utilized 2048 TPU cores for 4 days. This model achieves the best translation quality in terms of average BLEU, but also takes a total of 22.4 TPU years to train. While we have not seen any signs that the quality improvements plateau as we scale up our models, we strive for finding cost-effective solutions for

---

[3]Training loss reported in this section corresponds to cross-entropy loss and excludes the auxiliary loss term introduced in Section 2.1

[4]TPU core years is simply measured by the product of number of cores and wall-clock time in years.

scaling. Results in Table 1 again validates scaling with conditional computation is way more practical compared to dense scaling. Given the same number of TPU cores used by MoE(2048E, 36L), the dense scaling variant, T(96L), appears to be taking more than ten times to train (235 TPU core years), while trailing behind in terms of model quality compared to models trained with GShard.

## 4  RELATED WORK

**Model parallelism** partitions computation of neural network to build very large models on a cluster of accelerators. For example, pipelining (Huang et al., 2019; Harlap et al., 2018) splits a large model's layers into multiple stages, while operator-level partitioning (Shazeer et al., 2018; Jia et al., 2019) splits individual operators into smaller parallel operators. GShard used a type of operator-level partitioning to scale our model. Without the need to rewrite the model implementation on other frameworks, GShard only requires users to annotate how tensors are split on existing model code, while not worrying the correct reduction and data exchange over partitions, because that is handled by the compiler. GShard solved many practical problems when implementing SPMD transformation on a production compiler (XLA). For example, to our knowledge, it is the first work showing how we can partition unevenly-shaped, non-trivial ops that have spatial dimensions with complex static configurations (e.g., convolutions with static dilation and padding).

**Conditional Computation** Conditional computation (Bengio et al., 2015; Elbayad et al., 2020) postulates that examples should be routed within the network by activating an input dependent sub-network. Prior work (Bapna et al., 2020; Yang et al., 2019; Shazeer et al., 2017) have shown its promising applications in machine translation, language models and computer vision. The routing strategy can be any of the following: estimated difficulty of the example (Lugosch et al., 2020), available computation budget (Elbayad et al., 2020; Bapna et al., 2020), or more generally a learned criterion with sparsity induced mixture of experts (Shazeer et al., 2017). This paper extended sparsely gated mixture of experts to Transformers (Vaswani et al., 2017) and introduced novel gating function with efficient implementation on parallel devices.

**Model scaling** Within a single model family, simply making the network wider or deeper often improves the model quality empirically. E.g., deeper ResNets performed better (He et al., 2016b), bigger Transformer models achieved better translation quality (Vaswani et al., 2017), models with larger vocabulary, or embedding or feature crosses work better, too (Arivazhagan et al., 2019; Conneau et al., 2019). Across different model families, it has also been observed that bigger models with larger model capacities not only fit the training data better but also generalize better on test time (Zhang et al., 2017; Neyshabur et al., 2017; Huang et al., 2019). This observation motivated many research efforts to build much bigger neural networks than those typically used in deep learning research models or production models. Shazeer et al. (2017) showed that a recurrent language model with 69 billion parameters using mixture-of-expert layers achieved much lower test perplexity for the one billion words (LM1B) benchmark. Brown et al. (2020a) showed that a dense 175 billion parameters model is capable of exhibiting highly accurate few-shot performance on downstream NLP tasks.

## 5  CONCLUSION

Our results in this paper suggest that progressive scaling of neural networks yield consistent quality gains, validating that the quality improvements have not yet plateaued as we scale up our models. We applied GShard, a deep learning module that partitions computation at scale automatically, to scale up MoE Transformer with light weight sharding annotations in the model code. We demonstrated a 600B parameter multilingual neural machine translation model can efficiently be trained in 4 days achieving superior performance and quality compared to prior art when translating 100 languages to English with a single model. MoE Transformer models trained with GShard also excel at *training efficiency*, with a training cost of 22 TPU v3 core years compared to 29 TPU years used for training all 100 bilingual Transformer baseline models. Empirical results presented in this paper confirmed that scaling models by utilizing conditional computation not only improve the quality of real-world machine learning applications but also remained practical and sample efficient during training. Our proposed method presents a favorable scalability/cost trade-off and alleviates the need for model-specific frameworks or tools for scaling giant neural networks.

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

## A  APPENDIX

### A.1  RELATED WORK

**Neural networks** Deep learning models have been very successful in advancing sub-fields of artificial intelligence. For years, the fields have been continuously reporting new state of the art results using varieties of model architectures for computer vision tasks (Krizhevsky et al., 2012; Szegedy et al., 2015; He et al., 2016a), for natural language understanding tasks (Sutskever et al., 2014; Bahdanau et al., 2014; Wu et al., 2016), for speech recognition and synthesis tasks (Hinton et al., 2012; Chan et al., 2016; Chiu et al., 2018; Oord et al., 2016; Shen et al., 2018). More recently, attention-based Transformer models further advanced state of the art of these fields (Vaswani et al., 2017; Devlin et al., 2018; Shen et al., 2019).

**Hardware** Neural networks demand non-negligible amounts of computation power. To address such a demand, special hardware (chips and networked machines) built for neural network training and inference can be dated back to 25 years ago (Ienne et al., 1996). Since late 2000s, researchers started to leverage GPUs to accelerate neural nets (Raina et al., 2009; Krizhevsky et al., 2012; Cireşan et al., 2010). More recently, the industry also invested heavily in building more dedicated hardware systems chasing for more cost-effective neural network hardware (Jouppi et al., 2017). Because the core computation of neural networks (various forms of summation of multiplications: convolution, matrix multiplication, einsum) are highly parallelizable numerical calculations, these chips are equipped with huge number of floating processing units (FPUs). Hence, the compute power of these specially designed hardware grew dramatically. It is reported that GPU price per flops dropped a factor of ten in just the last 4 years (gpu) and flops per watts increased by 2 magnitude over the past 12 years (Sun et al., 2019). The widely available low-cost computation power is a major enabler for the success of neural networks.

**Software** Software systems supporting neural networks evolved together with the advancement of the underlying hardware (Dean et al., 2012; Bastien et al., 2012; Abadi et al., 2016; Paszke et al., 2017; Palkar & Zaharia, 2019). While the accelerators are highly parallel compute machines, they are significantly more difficult to program directly. The frameworks made building neural networks easier and abstracted away many hardware specific details from the practitioners. They in turn rely on lower-level libraries to drive special hardware (accelerators) efficiently. E.g., CUDA (Nickolls et al., 2008) for Nvidia's GPUs, or XLA for Google's TPUs (xla, 2019). These lower-level libraries are critical for achieving high efficiency using these special hardware.

**Automated parallelism** Because programming in a distributed heterogeneous environment is challenging, particularly for high-level practitioners, deep-learning frameworks attempt to alleviate the

burden of their users from specifying how the distributed computation is done. For example, Tensor-Flow (Abadi et al., 2016) has support for data parallelism, and basic model parallelism with graph partitioning by per-node device assignment. Mesh TensorFlow (Shazeer et al., 2018) helps the user to build large models with SPMD-style per-operator partitioning, by rewriting the computation in a Python library on top of TensorFlow; in comparison, our approach partitions the graph in the compiler based on light-weight annotations without requiring the user to rewrite the model. FlexFlow (Jia et al., 2019) uses automated search to discover the optimal partition of operators in a graph for better performance; while it focuses on determining the partitioning policy, our SPMD partitioner focuses on the mechanisms to transform an annotated graph. Weight-update sharding (Xu et al., 2020) is another automatic parallelization transformation based on XLA, which mostly focuses on performance optimizations for TPU clusters, and conceptually can be viewed as a special case for GShard. Zero (Rajbhandari et al., 2019) presents a set of optimizations to reduce memory redundancy in parallel training devices, by partitioning weights, activations, and optimizer state separately, and it is able to scale models to 170 billion parameters; in comparison, GShard is more general in the sense that it does not distinguish these tensors, and all of those specific partitioning techniques can be supported by simply annotating the corresponding tensors, allowing us to scale to over 1 trillion parameters and explore more design choices.

## A.2 THE XLA SPMD PARTITIONER FOR GSHARD

This section describes the compiler infrastructure that automatically partitions a computation graph based on sharding annotations. Sharding annotations inform the compiler about how each tensor should be distributed across devices. The SPMD (Single Program Multiple Data) partitioner (or "partitioner" for simplicity) is a compiler component that transforms a computation graph into a single program to be executed on all devices in parallel. This makes the compilation time near constant regardless of the number of partitions, which allows us to scale to thousands of partitions. [5]

We implemented the partitioner in the XLA compiler xla (2019). Multiple frontend frameworks including TensorFlow, JAX, PyTorch and Julia already have lowering logic to transform their graph representation to XLA HLO graph. XLA also has a much smaller set of operators compared to popular frontend frameworks like TensorFlow, which reduces the burden of implementing a partitioner without harming generality, because the existing lowering from frontends performs the heavy-lifting to make it expressive. Although we developed the infrastructure in XLA, the techniques we describe here can be applied to intermediate representations in other machine learning frameworks (e.g., ONNX onn (2019), TVM Relay Roesch et al. (2018), Glow IR Rotem et al. (2018)).

XLA models a computation as a dataflow graph where nodes are operators and edges are tensors flowing between operators. The core of the partitioner is per-operation handling that transforms a full-sized operator into a partition-sized operator according to the sharding specified on the input and output. When a computation is partitioned, various patterns of cross-device data transfers are introduced. In order to maximize the performance at large scale, it is essential to define a core set of communication primitives and optimize those for the target platform.

### A.2.1 SHARDING PROPAGATION

GShard only requires the user to annotate a few key tensors in the model, and the compiler will propagate them to all tensors on the graph in an optimization pass. This allows the user to reuse legacy model code by adding a small set of annotations.

The propagation pass is designed to be intuitive, and it mostly passes through shardings along shared dimensions between inputs and outputs. Typically, it requires annotations on model weights, and if sharding involves multiple dimensions, activations could also be annotated around core computation operators like Einsum which could have multiple possible outcomes of sharding propagation.

### A.2.2 PER-OPERATOR SPMD PARTITIONING

The core of the partitioner is the per-operator transformation from a full-sized operator into a partition-sized operator according to the specified sharding. While some operators (e.g., elementwise)

---

[5]An alternative is MPMD (Multiple Program Multiple Data), which does not scale as shown in Figure 4.

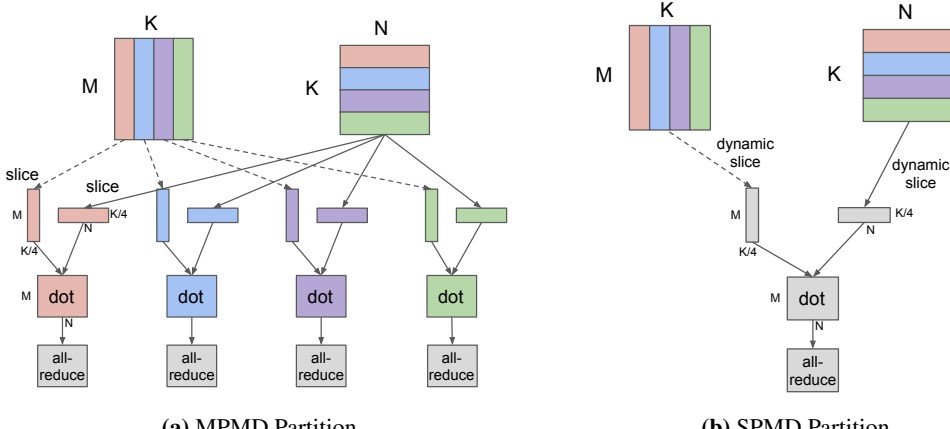

**(a)** MPMD Partition  **(b)** SPMD Partition

**Figure 4:** Comparison between MPMD and our proposed SPMD partitioning of a Dot operator ($[M, K] \times [K, N] = [M, N]$) across 4 devices. In this example, both operands are partitioned along the contracting dimension $K$, where each device computes the local result and globally combines with an AllReduce. MPMD partitioning generates separate operators for each device, limiting its scalability, whereas SPMD partitioning generates one program to run on all devices. Note that the compilation time with our SPMD partitioning is not-dependent of the number of devices being used.

are trivial to support, we discuss several common cases where cross-partition communications are required.

To keep the discussion more relevant to the MoE model, this section focuses on *Einsum* partitioning to illustrate a few communication patterns. And to keep it simple for now, we assume that all tensors are evenly partitioned, which means the size of the dimension to partitition is a multiple of the partition count.

**Einsum Case Study** *Einsum* is the most critical operator in implementing the MoE model. They are represented as a *Dot* operation in XLA HLO, where each operand (LHS or RHS) consists of three types of dimensions:

- **Batch dimensions** are the embarrassingly parallel dimensions. The same set of batch dimensions must exist in all of LHS, RHS and the output, and each element in the output only depends on the corresponding batch in LHS and RHS.

- **Contracting dimensions** only exist in the operands. LHS and RHS must have the same set of contracting dimensions, and they are summed up and collapsed in the output.

- **Non-contracting dimensions** are also parallel dimensions that exist in one of the operands and the output. Each of LHS and RHS has its own set of non-contracting dimensions, which are inherited by the output.

Sharding propagation prioritizes choosing the same sharding on batch dimensions of LHS, RHS and output, because that would avoid any cross-partition communication. However, that is not always possible, and we need cross-partition communication in the following three cases.

- **Resharding.** In the MoE model we built, the expert dispatching logic (Line 3 in Algorithm 2) requires switching the partitioned dimension after an *Einsum*. Since resharding is efficient (Section A.3.2) with *AllToAll*, we first execute the *Einsum* locally, then reshard it to the desired dimension, as shown in Figure 5a.

- **Accumulating partial results.** If the inputs are partitioned along contracting dimensions, the local result is partial and we need to use an *AllReduce* to combine them and produce the final result, as shown in Figure 5b.

- **Slicing in a loop.** For certain scenarios, we also implemented an algorithm similar to Cannon's algorithm Cannon (1969), in order to limit the size of tensors on each partition. For example, if both operands are partitioned on a non-contracting dimension, we cannot compute the local *Einsum* directly since operands have different non-contracting dimensions.

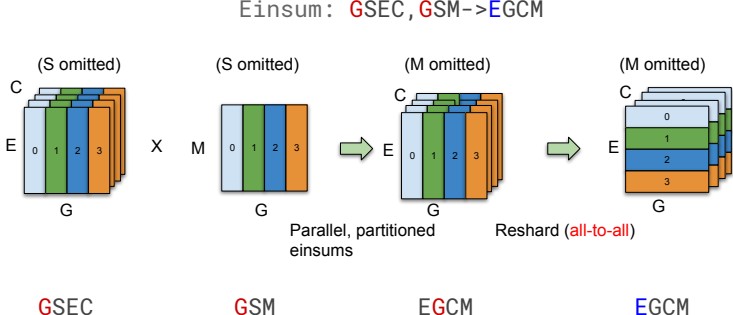

**(a)** A partitioned *Einsum* operator. Colored letters ($G$ and $E$) represent the partitioned dimension of each tensor. The partitioner decides to first execute a batch-parallel *Einsum* along the $G$ dimension, then reshard the result to the $E$ dimension.

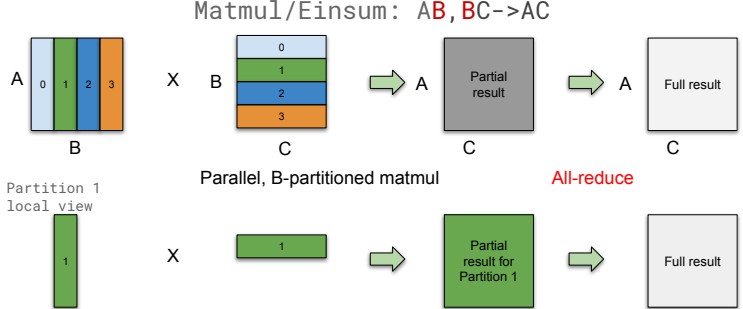

**(b)** A simple *Einsum* (*Matmul*) partitioned on the contracting dimension.

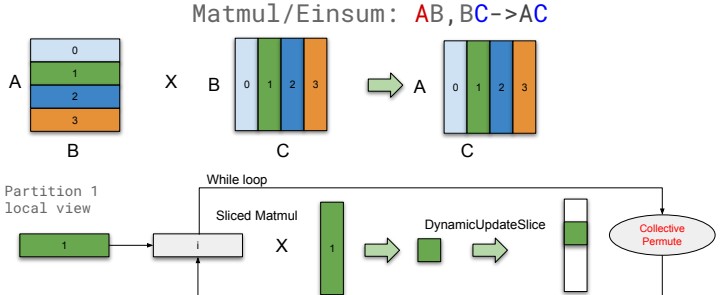

**(c)** An *Einsum* (*Matmul*) where we use collective-permute in a loop to compute one slice at a time. There is no full-sized tensor during the entire process.

**Figure 5:** Examples of *Einsum* partitioning with cross-device communication.

Replicating one of the operands would not cause redundant computation, but it requires the replicated operand to fit in device memory. Therefore, if the size of the operand is too large, we instead keep both operands partitioned and use a loop to iterate over each slice of the result, and use *CollectivePermute* to communicate the input slices (Figure 5c).

**Compiler optimizations** The SPMD partitioner creates various data formatting operators in order to perform slicing, padding, concatenation, masking and halo exchange. To address the issue, we leverage XLA's fusion capabilities on TPU, as well as code motion optimizations for slicing and padding, to largely hide the overhead of data formatting. As a result, the run-time overhead is typically negligible, even for convolutional networks where masking and padding are heavily used.

### A.2.3 General Sharding API

In addition to the two common APIs (*replicate()* and *split()*) for sharding listed in Section 2.2, users or the compiler may use a more advanced sharding strategy to minimize data transfers.

**shard(tensor, device_assignment)** annotates *tensor* to be partitioned with the provided device assignment, and returns the annotated tensor. We use *device assignment*, a multi-dimensional integer array, to represent how the split is done. *device_assignment* has the same rank as the data tensor; its element count is the total number of partitions, and each element is the ID of the device that occupies the corresponding data slice. For example, a 3D tensor with shape $[256, 1024, 8192]$ with device assignment shape $[2, 1, 4]$ will have partition shape $[128, 1024, 2048]$, and the order of elements in *device assignment* determines which slice each partition occupies.

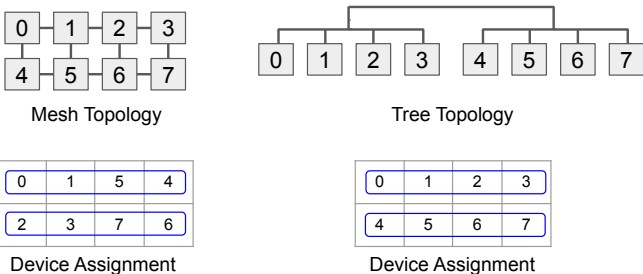

**Figure 6:** An example of two different *device assignments* based on the device topology. A 2D tensor is split by 2x4 partitions and the communication pattern is between partitions along the rows of the tensor. The numbers represent device ids.

Since data movement across devices critically affects the parallel execution performance, it is important to consider the target device topology as well as the communication between partitions of the tensor when assigning device ids in the *device assignment* for maximum performance. Figure 6 shows two different *device assignments* based on the device topology and the row-wise communication pattern on the tensor.

### A.3 Performance and Memory Consumption

This section discusses how well GShard achieves computation and memory efficiency on the TPU platform. Our measurement and analysis show that the device memory consumption is roughly constant when we increase the number of devices and experts, and the step time grows sublinearly, i.e., 1.7x execution time increase when we scale the model by 16x from 128 devices to 2048 devices. We also provide microbenchmarks and analyses for a variety of partitioned operators, which could guide use cases beyond this paper.

### A.3.1 Memory Efficiency and Scalability

In the GShard model, there are mainly three types of memory usage, all of which have constant per-device sizes after SPMD partitioning, when the number of experts increases.

- Replicated weights (e.g. transformer feed-forward layers).
- Distributed weights (MoE feed-forward layers[6]).
- Activations (output of each layer that is used in both forward and backward pass).

The $O(1)$ memory scaling is demonstrated in Figure 7, which shows the per-device memory usage distribution for different models. With a fixed number of layers, both weight memory and activation memory stay constant when the number of experts increases.

On this other hand, weight memory and activation memory both scale linearly with the number of layers. When the memory requirement exceeds available memory on each device, compiler-based

---

[6]Gate projection weights are $O(E)$ in size and could be partitioned, but in practice they are small enough to be replicated and only have negligible effect on peak memory usage.

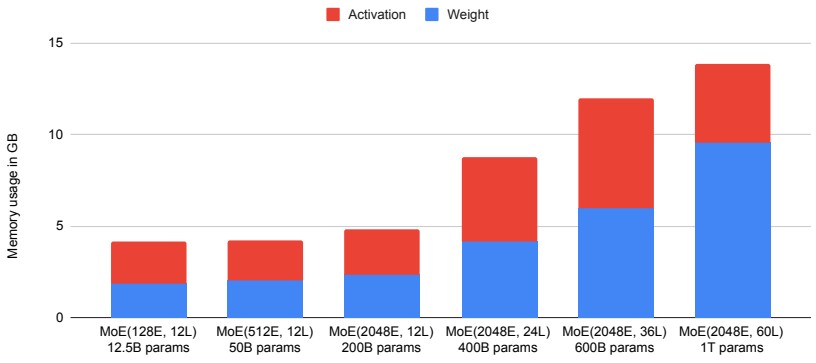

**Figure 7:** Per-device memory consumption in gigabytes.

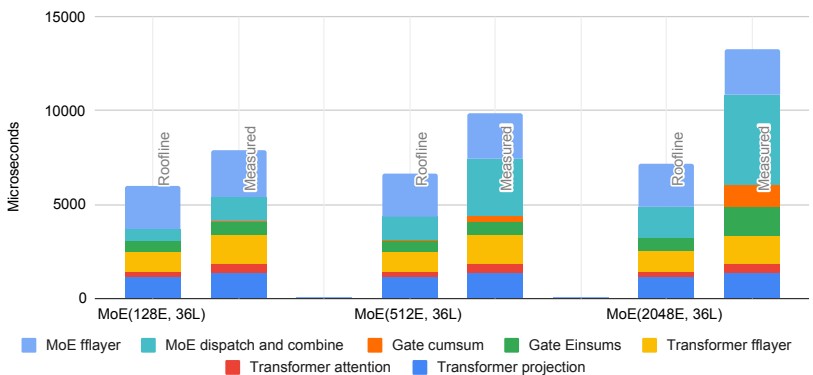

**Figure 8:** Measured vs roofline execution time breakdown. Only the forward pass is shown, and the backward pass has similar breakdown. "MoE dispatch and combine" represents cross-partition communication with *AllToAll*.

rematerialization will automatically recompute part of the activations in the backward pass in order to reduce peak activation memory. This is why the activation size for MoE(2048E, 60L) is smaller than MoE(2048E, 36L). The overhead of rematerialization is also optimized, e.g. only 28% and 34% of the total cycles are spent on recomputation for 36L and 60L models respectively, and 0% for 12L and 24L since they fit in device memory without rematerialization.

### A.3.2 RUNTIME EFFICIENCY AND SCALABILITY

Figure 8 shows the breakdown of execution time for an MoE layer and its adjacent Transformer layer. It also compares the achieved performance to a roofline, which is estimated by assuming compute-, memory-, or communication-bounded operations can achieve 100% of the peak FLOPS, memory bandwidth, or interconnect bandwidth. This is a very optimistic estimate as many operators are bounded by a mixed set of resources. At a smaller scale (128 experts), our model can achieve > 70% of the roofline performance. The device time increases by 1.7x when we scale the model to 16x larger (2048 experts), and can still achieve 48% of the roofline performance.

**Transformer layers and MoE feed-forward layer** These are the dense parts of the model, which are designed to achieve peak TPU utilization. On each device, these computations also have a constant cost when we scale to more experts. Feed-forward layers and Transformer projections are mainly large matrix multiplications that utilize the TPU's matrix unit well. These operations have achieved > 85% peak FLOPS in our experiment. The attention operations are composed of mainly batch matmuls, which are bounded by memory bandwidth when sequence lengths are small. As a result, in our experiments attention operations only achieved > 30% peak FLOPS.

**Gate computation** In Figure 8, "Gate Einsum" represents the first two and the last *Einsums* in Algorithm 2. The first *Einsum* is the projection that calculates per-expert input to *softmax*. It has an $O(D)$ cost, but it is a very small part of the layer. The other two *Einsums* are dispatching tokens and

combining expert results. They effectively implement *Gather* with one-hot matrices, which are more expensive, but with constant $O(GC) = O(1)$ cost that is independent from the number of experts. The execution time of these *Einsums* increases by around 2x when we scale from 128 to 2048 experts (16x).

The remaining per-device gating computation involves many general-purpose computations like *ArgMax* and *Cumsum*, which are either memory-bound or even sequential in nature, thus not designed to utilize TPUs well. The majority of the time is spent on sequential *Cumsum* operations to invert one-hot matrices that represent selected experts for each token to one-hot matrices that represent selected tokens for each expert. The linear complexity of *Cumsum* is demonstrated in Figure 8. This part of the gating computation also has an $O(D)$ cost, but fortunately, similar to the *Einsum* before *softmax*, it has a very small constant factor. It has negligible execution time with 128 experts, and takes less than 10% of the total time spent in the MoE and Transformer layers with 2048 experts.

The most significant part of gating is communication, shown as "MoE dispatch and combine" in Figure 8. These are *AllToAll* operators, and as we will discuss in Section A.3.3, their cost is $O(\sqrt{D})$. When the number experts grows 16x from 128 to 2048, the execution time increases by about 3.75x, and their proportion of execution time in the MoE and Transformer increases from 16% to 36%.

### A.3.3 COMMUNICATION MICROBENCHMARKS AND PER-OPERATOR SCALABILITY

In this section, we measure and analyze the performance scalability of the SPMD partitioner for basic operators, which can be used to guide use cases beyond the MoE model presented in this paper.

**Performance scaling of communication primitives** Two critical collective communication operators in the MoE model are *AllReduce* and *AllToAll*. *AllReduce* is used in accumulating partial results, and *AllToAll* is used in resharding (Section A.2.2). Figure 9 shows their performance scalability from 16 to 2048 partitions. *AllReduce* on TPU has an execution time independent from the number of devices Ying et al. (2018). The variance in Figure 9 is due to specifics of each topology, e.g., whether it is a square or a rectangle, and whether it is a torus or a mesh.

*AllToAll*, on the other hand, gets more expensive as the number of partitions grows, but in a sublinear manner. On our 2D TPU cluster, *AllToAll* cost is roughly $O(\sqrt{D})$, where $D$ is the number of partitions. This is because with a fixed amount of data each partition sends (8MB or 32MB in Figure 9), the total amount of data that all partitions send is $d = O(D)$. Meanwhile, each data piece needs to travel $h = O(\sqrt{D})$ hops on average, and there are overall $l = O(D)$ device-to-device links in the network. Therefore, if it is bandwidth-bound, the execution time of an *AllToAll* is

$$t = \frac{dh}{l} = O(\frac{D\sqrt{D}}{D}) = O(\sqrt{D}).$$

Even if it is latency-bound, the execution time will still be $O(h) = O(\sqrt{D})$. Comparing 2048 partitions and 16 partitions, while $D$ grows by 128 times, the execution time of *AllToAll* only increases by 9 times. This enables us to use resharding to efficiently implement cross-partition dispatching (Figure 5a).

*AllGather* and *CollectivePermute* are easier to analyze. *AllGather*'s output is $D$ larger than the input, and if we fix input size, then its communication cost is $O(D)$. *CollectivePermute* has a one-to-one communication pattern, and with reasonable device arrangement where the source-destination pairs are close, its cost is $O(1)$ for a fixed input size.

**Partitioned operator scalability** We summarize the performance scalability for common operators using GShard in Table 2. It contains the *Einsum/Matmul* examples in Section A.2.2, and also other common operators like *Convolution* and *Reduce*. The table includes the local compute on each partition, as well as the required communication based on our analysis above.

Most operators in Table 2 have sublinear scalability in terms of both compute and communication, which is consistent with our performance measurement of the MoE model. The $O(1)$ scaling of spatially partitioned convolutions also demonstrates the efficiency of GShard for image partitioning.

However, the last two *Matmul* operators in Table 2 have $O(D)$ scaling of per-partition compute and communication, where they have unmatched sharding in the operands. This is not due to inefficiency in the partitioning algorithm, but because the total compute in the full operator is very large ($O(D^2)$).

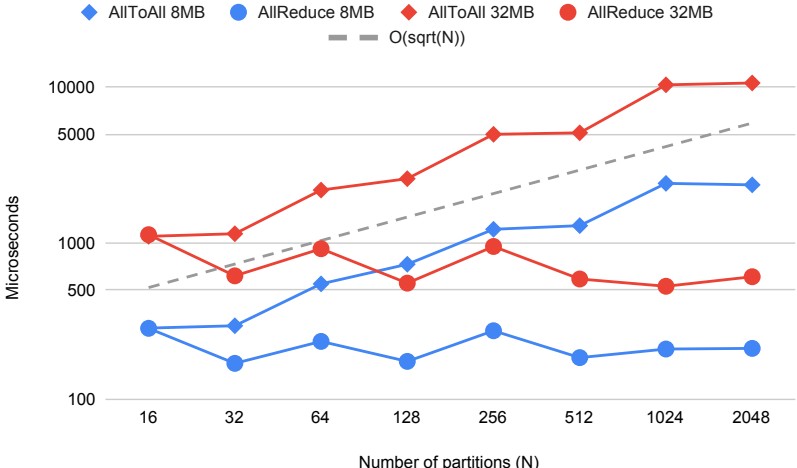

**Figure 9:** Performance scaling of communication, *AllReduce* and *AllToAll*. Log scale on both axes. *AllReduce* cost is roughly $O(1)$, and *AllToAll* cost is roughly $O(\sqrt{D})$, where $D$ is the number of partitions. We measure their performance with 8MB and 32MB data. For *AllToAll*, that means each partition initially has 8MB (or 32MB) data, then divides it to $D$ pieces, and sends each piece to a different receiving partition.

|  | $O(D)$ Dimensions | Total Compute | Per-partition Compute | Communication |
|---|---|---|---|---|
| *Add($\underline{A}$,$\underline{A}$->$\underline{A}$)* | $A$ | $O(D)$ | $O(1)$ | 0 |
| *Matmul($A\underline{B}$,$\underline{B}C$->$AC$)* | $B$ | $O(D)$ | $O(1)$ | $O(1)$ *AR* |
| *Matmul($\underline{A}B$,$BC$->$\underline{A}C$)* | $A$ | $O(D)$ | $O(1)$ | 0 |
| *Matmul($A\underline{B}$,$\underline{B}C$->$AC$)* | $A,B$ | $O(D^2)$ | $O(D)$ | $O(D)$ *AG* or *CP* |
| *Matmul($\underline{A}B$,$B\underline{C}$->$A\underline{C}$)* | $A,C$ | $O(D^2)$ | $O(D)$ | $O(D)$ *AG* or *CP* |
| *Reduce($\underline{A}B$->$\underline{A}$)* | $A$ | $O(D)$ | $O(1)$ | 0 |
| *Reduce($\underline{A}B$->$B$)* | $A$ | $O(D)$ | $O(1)$ | $O(1)$ *AR* |
| *Einsum($\underline{G}SEC$,$\underline{G}SM$->$\underline{E}GCM$)* | $G,E$ * | $O(D)$ | $O(1)$ | $O(\sqrt{D})$ *AA* |
| *Convolution($BI\underline{X}Y$,$xyIO$->$BO\underline{X}Y$)* | $X$ ** | $O(D)$ | $O(1)$ | $O(1)$ *CP* |

**Table 2:** Scalability of partitioned operators. Abbreviation for communication primitives: *AR: AllReduce*, *AG: AllGather*, *CP: CollectivePermute*, *AA: AllToAll*. *This is the dispatch *Einsum* in our model, where we set $C$ to $O(1/D)$. **I/O are the input/output feature dimensions, $B$ is the batch dimension, $X/Y$ are input spatial dimensions, and $x/y$ are the kernal spatial dimensions.

Different partitioning strategies can be used for these cases, producing different communication primitives: replicating one operand will result in *AllGather* (requiring the replicated operand to fit in device memory), while slicing in a loop (Figure 5c) will result in *CollectivePermute*.

## A.4 DENSE MODEL SCALABILITY AND BENCHMARKS

GShard is not limited to sparse models. In this subsection we applied GShard to build large dense transformer with up to trillions of parameters. We open sourced our example implementation and provided a step by step instruction how to train it on the public cloud provider (`https://github.com/tensorflow/lingvo/tree/master/lingvo/tasks/lm`). We included the model details and performance benchmarks in Table 3. To the best of our knowledge, we provide the only open source implementation that can train transformer models with trillions of parameters efficiently on public cloud. GShard allows tensor partitioning with more than one dimension. For example, we split the activation tensors along both the batch and the model dimensions. This allows input batches with long sequence length (1024 in the benchmark) and global batch size smaller than the number of devices. The performance scales linearly from 64B to 1T. The communication bottleneck started to dominate when scaling further to 4T as compute/communication ratio is lower due to small batch size. Larger batch size is possible by scaling out the model to more TPU cores, or by enabling

gradient accumulation. For the purpose of apple-to-apple comparison to other models, we did not include the above optimizations in the 4T model.

| # of params | # of layers | model dim | hidden dim | seq length | batch size | MXU util | k-tokens /sec |
|---|---|---|---|---|---|---|---|
| 64B | 32 | 8192 | 65536 | 1024 | 512 | 48.4% | 150.7 |
| 128B | 64 | 8192 | 65536 | 1024 | 512 | 48.5% | 73.5 |
| 1T | 128 | 16384 | 131072 | 1024 | 128 | 47.2% | 9.23 |
| 4T | 128 | 32768 | 262144 | 1024 | 32 | 28.8% | 1.36 |

**Table 3:** Dense language models benchmarks on TPU v3-2048. The model parameters are sharded across 2048 TPU V3 cores. We measured the TPU matrix unit utilization and throughput for dense models with various configurations.

**Details of dense Transformer sharding** We use a 2D mesh of TPU devices of shape [X, Y]. X is used to shard the batch dimension of activation tensors, and Y is used to shard attention heads and the feed-forward hidden dimension. Activations' head and hidden dimensions are sharded the same way along Y. To further reduce weight storage on each device, we additionally shard the model dimension of the weights along the X dimension. Weights will be partially unsharded on-demand with a subgrouped AllGather operator across devices along X. The parallelism pattern along X is conceptually equivalent to weight-update sharding Xu et al. (2020).

When we further increase the model size, activation storage becomes the bottleneck, because activations between transformer layers are only partially sharded in the device mesh on the batch dimension. So we further shard the model dimension of these activation tensors along Y, making them fully sharded across all devices. Such an activation tensor will be produced by a ReduceScatter operator, which is semantically AllReduce followed by a DynamicSlice but can be implemented more efficiently. The fully sharded activations will also be partially unsharded with AllGather in the next layer.

## A.5  DECODING WITH FLAT BEAM SEARCH

During decoding, we use beam search with length normalization similar to Wu et al. (2016). Decoding is auto-regressive and generates the target sequence one token at a time, so for an output of length $m$ the decoder layer stack is executed $m$ times, sequentially. In particular for each decoder MoE layer there are dispatch/combine operations, which require cross-device communication. Inference utilizes same cluster with same number of devices as training.

During beam search we flatten the beam hypotheses into a single sequence which contains all underlying tokens interleaved, and we modify decoder self-attention mask so that each hypothesis only has attention to appropriate positions in the joint flat sequence. We apply the same transformation to key/value tensors maintained by each decoder self-attention layer. This allows us to avoid reordering previously computed attention key/values after each beam expansion. Instead, we only reorder the $0/1$ mask representing the current active hypotheses. However, attention becomes $k$ times longer.

This trade-off can be positive or negative depending on implementation details. As explained in Shazeer (2019), memory bandwidth limits are important for incremental decoding with Transformer models. From this point of view, by flattening the beam we replace two operations with low compute/memory ratio (attention dot product and key/value reordering) with a single operation with a slightly higher compute/memory ratio (attention dot product over a longer sequence with more keys), but with the same total amount of memory it has to access.

## A.6  MACHINE TRANSLATION EXPERIMENTS DETAILS

In our Machine Translation experiments MoE Transformer models shared

- Transformer model dimension $M = 1024$
- Feed Forward and MoE hidden dimension $H = 8192$
- Number of heads in multi-head attention = 16
- Attention key and value dimension = 128

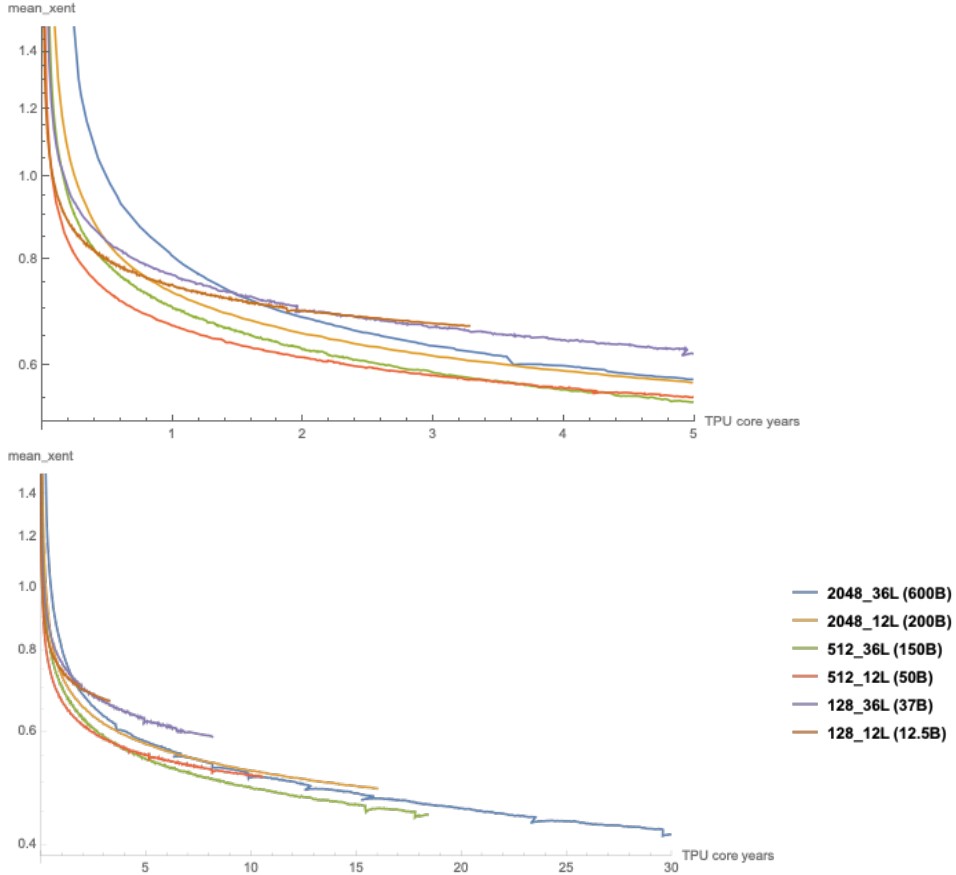

**Figure 10:** Training loss curves from different model scales under low (upper graph) and high (lower graph) compute budget. Lower loss can be obtained by growing the model capacity until some budget-dependent maximum size, which increased with higher training budget. Given the same five TPU core years training budget, we observed that the 150B model hits the lowest training loss. But with 30 TPU core years, 600B achieved the lowest loss.

- Input, residual and attention dropout rate = 0.1
- The number of groups $G = 2D$, twice the number of devices.
- The expert capacity $C = 2 \propto \frac{B \times L}{D \times E}$.

.

We used the Adafactor (Shazeer & Stern, 2018) optimizer with *a)* factored second-moment estimation; *b)* first moment decay $\beta_1 = 0.0$; *c)* second moment decay $\beta_2 = 0.99$ with $1 - t^{-0.8}$ schedule; *d)* clipping threshold of 1.0; and *e)* 1.0 learning rate with square root decay after 10k training steps.

We used SentencePiece Kudo & Richardson (2018) subword tokenizer with a single multilingual vocabulary for source-side spanning 102 languages of size 64000, and English-only target-side vocabulary of size 32000.

In Figure 10, we compare the achieved loss of each model at different preset training budgets. We observed that lower loss can be obtained by growing the model capacity until some budget-dependent maximum size, which increased with higher training budget. For example, with a relatively low training budget of 5 TPU core years training budget, we observed models with larger capacity lead to even lower training loss up to 150B parameters. But with a high training budget of 30 TPU core years, 600B achieved the lower cost.

