# OpenReview forum: "GShard: Scaling Giant Models with Conditional Computation and Automatic Sharding"
_ICLR.cc/2021/Conference — ICLR 2021 Poster_

### Official Review · AnonReviewer2 · 2020-10-17
**The paper demonstrates a neural architecture that combines transformers with mixture-of-experts, and shows that it can be trained on up to 600 billion parameters, a roughly 5-10 fold increase over pure transformer architectures. The increased number of parameters allows improved BLEU scores on a large-sclae multilingual translation task.**

**Rating:** 4
**Confidence:** 4

**Review:**

Claims:

-Architecture based on transformer + mixture of experts

-Number of parameters can scale 5-10x larger than transformer-only architectures, while simultaneously achieving 10x shorter training time

-Increased number of parameters allows improvements in BLEU score

-MoE architecture allows sub-linear computational scaling with increasing parameters, even when mixed with transformer layers

-Argues that conditional computation, via the GShard module, is a good way to tackle computation cost and complexity of parallel programming

Pros:

-600 billion parameters is - as far as I am aware - a new record for the size of transformer-related models.

-It is very promising to see that increasing the number of parameters (via the density of the transformer layers, or the depth of the model), continues to yield BLEU score increases on very large datasets.

-Decreasing computational cost and programming complexity is an important goal - however, I think this should be studied not just at the rarefied high end of extremely large models and datasets, but on more typical scales that would be accessible to more research teams.

-The paper is quite thorough about implementation details, and comes with a comprehensive appendix. This will greatly help with reproducibility. However, I have concerns that the huge size of the sole dataset/task unfortunately work in the opposite direction, hurting reproducibility

Cons:

-The paper only studies one task, on one data set, at one (very large) scale, and only compares to one baseline (standard transformer architecture). Although the paper deserves interest due to its potentially record-setting model size and scale, due to the lack of variety in tasks, data and scale, it comes across as a singular engineering effort rather than a study into the benefits of conditional computation. This creates two issues: (1) it is unclear how well the model generalizes to other tasks (does it only work for really big problems? or will it also be competitive for smaller problems); (2) even if the dataset were made public, would anyone be able to reproduce the results given that 23 TPU core-years is an impractically-high cost for most teams?

-The theme of conditional computation is interesting, but the MoE design isn't too much different from Shazeer et al., 2017. It was a little disappointing that different variations of conditional computation were not explored, perhaps through ablation studies.

-GShard is mentioned as a contribution, but the API design, particularly the ability to specify a dimension of parallelism, is not novel. My recollection is that Jia et al., SysML’2019 already covers the claims the authors are trying to make.

-Regarding the claims on page 7 about model quality improving with greater model depth and denser transformer layers - while intuitive, these claims are based on rather shallow evidence that compares two results points at a time. Consequently, the argument about "increased bandwidth for transfer" seems speculative, and deserves more substantiation. I think a well-designed ablation study would really help back up these arguments.

-There isn't much discussion about related work and how the paper makes novel contributions. Beyond the 600b parameter scale, it seems the key original contribution is to fuse transformers with MOE, but I did have a hard time understanding what other contributions were significant and novel.

Recommendations for improvement:

1) Having experiments on smaller model sizes, particularly in the 10m-10b range, would help make the results reproducible by typical research/engineering teams. Smaller, publicly-available datasets (ideally in the billions of tokens, rather than trillions) would go a long way too. If, on top of that, the authors studied a second task (other than multilingual translation), I think the paper would be very solid.

2) Variations of the MoE architecture, or perhaps ablation studies, could really help to bring out the conditional computation theme, improve the novelty of the paper, as well as provide stronger evidence for the quality-gain claims on page 7. For example, the authors talk about Load Balancing, Auxiliary Loss, and Random Routing, which I understand are not present in the original MoE paper. Perhaps these could be a starting point for designing a set of ablation experiments.

3) It would be good to understand what is novel/improved about GShard, compared to existing systems-ML literature

---

> ### Author Response · Authors · 2020-11-24
> **We thank the reviewer for the suggestions and feedback.**
>
> We thank the reviewer for the suggestions and feedback.
>
> We adopted most of the reviewer’s recommendations in our revision. For example, we highlighted our architectural or algorithmic innovations such as local dispatching, auxiliary loss, and random routing in Section 2.1. We also included a section of related work in the main text.
>
> We apologized for the lack of systematic ablation studies and lack of applications to the other tasks. This is the result of the fact that model training at this scale is extremely costly.  While we did not perform comprehensive ablation studies, some algorithmic and hyperparameters choices made are the result of resolving trainability issues for specific runs. Such issues are transient and could occur after substantial training period, thus it is problematic to perform full ablation study. But we compared models with various scales and translation quality differences between the moe models and the dense models in the experiment section. We also pointed out in the related work that prior work (Bapna et al., 2020; Yang et al., 2019; Shazeer et al., 2017) have shown the promising applications of conditional computation in machine translation, language models and computer vision.
>
>
> In this research project and in this paper, we are pioneering scaling and enabling models with 10x larger size than the previous approaches. Therefore, we focus more on sharing what is possible and what are the bottlenecks with the community, rather than highlighting or claiming algorithmic novelty.   We believe that our contribution unlocks several interesting research questions that could be further studied, and with the open-sourced code, enabling more investigations.  We also believe that, although not directly replicable, the pathologies that exist in mixture-of-experts models can be replicated and studied at smaller scales, such as "rich-gets-richer", load-balance and transfer-dynamics of mixture models.
>
> Regarding related work in SysMl,  the sharding representation on tensors is similar to some prior work, e.g., MeshTensorFlow and Jia SysML19. However, the novelty of our work is to solve many practical problems when implementing SPMD transformation on a production compiler (XLA) that has static shape restrictions; this is large amount of work, and to our knowledge, it is the first work 1) enabling dense language models with trillions of parameters thanks to our general annotation API that supports tensor sharding with more than one dimensions 2)  showing how we can partition unevenly-shaped, non-trivial ops that have spatial dimensions with complex static configurations (e.g., convolutions with static dilation and padding).  In comparison, Jia SysML19 focuses on finding the best sharding configuration and does not discuss how sharded operators are implemented, and what ops are supported. Another important feature of GShard is backward compatibility. In comparison, MeshTensorFlow is a Python library that would require the model to be rewritten; but GShard can be conveniently integrated to existing model code by adding a few annotations.
>
> An additional comment about the novelty of our work; it is indeed arguable what stands for novel and what is not, as could be grounded to the reviewer guideline of ICLR “Strong points: is the submission clear, technically correct, experimentally rigorous, reproducible, does it present novel findings (e.g. theoretically, algorithmically, etc.)?” We believe that the proposed conditional computation model, combined with GShard API has allowed us to stress test impactful research direction (eg. Machine Translation) and study the scaling dynamics of giant neural networks, which would otherwise be impossible to explore, hence, we believe our findings are novel.
>
> Furthermore, while it may be seen as minor modifications or incremental algorithmic improvements, they enabled models with orders of magnitude more parameters.  The enhancements proposed by our paper are distilled from long experimental cycles and analysis, as we were relaxing the capacity bottleneck, we were at the same time facing trainability challenges one after another, and although might be seen as minor, these contributions were the key to address trainability challenges, and letting us push the scaling of neural networks to the limits.

---

### Official Review · AnonReviewer1 · 2020-10-23
**Great performance but lack novelty**

**Rating:** 5
**Confidence:** 5

**Review:**

Summary:

The paper applies mixture-of-experts (MoE) [1] to the Transformers to significantly increase the number of parameters in the model while keeping the total computational cost feasible. The main differences with [1] are: (1) Only choose 2 experts at each timestep; (2) Set a capacity upper bound for each expert to make sure that no expert becomes a lagger. The paper also presents a convenient library to make implementing MoE models easier. The proposed method is tested on a large multilingual machine translation dataset and shows performance gains over models trained on a single language pair and a model trained without MoE.

Reasons for score:

Although the improvement of translation performance presented in the paper is significant, my overall feeling is that the improvement over [1] is incremental and not clearly justified. See the following pros and cons for more detail.

Pros:
1. The improvements in translation accuracy are significant and show the effectiveness of MoE for Transformers. This should be an interesting result for the community and beneficial for people working on scaling Transformers.
2. If publicly released, the GShard library should be useful for implementing MoE or more general model parallel models.

Cons:
1. The novelty of the paper, especially compared with [1]. In [1], MoE is implemented as a position-wise feed-forward layer, which can be directly applied to Transformers with no modification. Also, there already exist previous works applying MoE on Transformers, such as [2].
2. The main improvement over [1] is to add a capacity bound on each expert to mitigate the lagger issue. However, this improvement is not thoroughly evaluated in the paper. I would expect an ablation study to measure the impact on the speed and accuracy of this improvement.
3. The paper proposed a library and a set of APIs for the XLA compiler to make the implementation easy. However, there are previous works (e.g. [3]) already working on a similar but more general set of APIs. How does this work differ from these works, especially [3]?
4. The performances on MT are great. However, all of the evaluations are not on any standard MT test sets (e.g. test sets of WMT). I understand it’s impossible to evaluate the performance of all 100 languages with WMT, but including the results of the standard benchmarks will make the performance gain claimed in the paper more intuitive and stronger.
5. For models of different sizes, it would be better to include a comparison of models training with a similar compute budget (e.g. training the models with the same TPU-days) to better show the benefit of large models.

Minor comments:
- At the bottom on page 2: “in sufficiently” -> “insufficiently”.
- Figure 2: There is an extra black solid line in the first multi-head attention layer on device E (at the bottom-right of the figure).
- Second paragraph from the bottom of page 3: “software Stack” -> “software stack”.
- Algorithm 2: Please define H and M.

References:

[1] Shazeer, Noam, et al. "Outrageously large neural networks: The sparsely-gated mixture-of-experts layer." arXiv preprint arXiv:1701.06538 (2017).

[2] Shen, Tianxiao, et al. "Mixture models for diverse machine translation: Tricks of the trade." arXiv preprint arXiv:1902.07816 (2019).

[3] Palkar, Shoumik, and Matei Zaharia. "Optimizing data-intensive computations in existing libraries with split annotations." Proceedings of the 27th ACM Symposium on Operating Systems Principles. 2019.

---

> ### Author Response · Authors · 2020-11-24
> **Appreciate the feedback and revised our paper accordingly.**
>
> We appreciate the constructive feedback and valuable suggestions.
>
> We open sourced our GShard implementation and provided an example implementation for the GPT-3 style dense transformer that can efficiently train language models with trillions of parameters and sequence length 1024. We included a table in Appendix A3.4 to show the throughput benchmarks. To the best of our knowledge,  we provided the only open source implementation that can train transformer models with trillions of parameters efficiently on public cloud.
>
> We also fixed the minor comments reviewer mentioned in the draft. And here are more point-by-point responses to reviewer’s feedback:
>
> 1) We focused on providing model details for reproducibility and didn’t do a good job at highlighting our algorithmic innovations in section 2.1.  [1]’s gating function is sequential, it limits the efficiency when scaling to thousands of devices. We introduced a new parallel gating function to get around this limitation. The proposed method in [2] is making use of mixture-of-experts as trainable ensemble models, and studies the governing variables for diversity in machine translation where we investigated the use of conditional computation for scaling translation models. The definition of "expert" in [2] is the entire translation model (akin to mixture of Transformers), where in our study, we defined experts as sub-networks. We agree that there are parallels between the two approaches, such as degeneracies in expert models which our paper also suffers from, such as "rich-gets-richer", which we will add to the discussion. We also want to note that the parametrization proposed by [2] introduces latent variables where we introduce gating sub-networks.
>
> 2) In general, adding capacity improves the machine translation quality since it reduced the number of overflowed tokens. We generally increase the capacity until the portion of overflowed tokens is low.  Since G * C = 2 \times # number of tokens in the batch, increasing the capacity leads to decrease in number of groups, which hurts the throughput by limiting the number of parallel gating. It’s a tradeoff between model accuracy and speed. And local dispatching enables O(G) speedup vs. without it.  We couldn’t do ablation study since training is costly so we picked the tradeoff at small scale and used it for all other scales.
>
> 3) Thanks for pointing out this related work. [3] is very different from GShard. First, it aims to improve single-machine performance of  multi-threads libraries, instead of networked devices, so they would not need collective communication operations; their goal is more similar to operator fusion. Second, [3] treat library functions as blackboxes, and require the developer to understand the semantics of these functions to add the correct annotations and reduction functions; instead, GShard only requires the user to annotate how tensors are split while not worrying about the correct reduction and data exchange over partitions, because that is handled by the compiler. We added [3] in the related work.
>
> 4) Wmt test data can not be directly used for evaluation since our training set was crawled from the web so it may have included the test data. Training on wmt data does not allow to study model scaling due to limited dataset size, so we had to resort to in-house training dataset to discover scaling laws with such big models
>
> 5) For models of different sizes, we concluded deeper models consumed less training samples to reach some present training loss thresholds. We included a comparison of models training with a similar compute budget in terms of both TPU core years and the number of training samples in Appendix A5. For example, the general trend we observe from Table 3 is that MoE Transformer models with 3 times the depth need 2 to 3 times fewer tokens to reach the preset training loss thresholds.  In Figure 14 of appendix A5, we compare the achieved loss of each model at different preset training budgets. We observed that lower loss can be obtained by growing the model capacity until some budget-dependent maximum size, which increased with higher training budget. For example, with a relatively low training budget of 5 TPU core years training budget, we observed models with larger capacity lead to even lower training loss up to  150B parameters. But with a high training budget of 30 TPU core years, 600B achieved the lower cost.

---

> > ### Author Response · Authors · 2020-11-24
> > **An additional comment about the novelty**
> >
> > An additional comment about the novelty of our work; it is indeed arguable what stands for novel and what is not, as could be grounded to the reviewer guideline of ICLR “Strong points: is the submission clear, technically correct, experimentally rigorous, reproducible, does it present novel findings (e.g. theoretically, algorithmically, etc.)?” We believe that the proposed conditional computation model, combined with GShard API has allowed us to stress test impactful research direction (eg. Machine Translation) and study the scaling dynamics of giant neural networks, which would otherwise be impossible to explore, hence, we believe our findings are novel.
> >
> > Furthermore, while it may be seen as minor modifications or incremental algorithmic improvements, they enabled models with orders of magnitude more parameters. For example,  to the best of our knowledge, GShard is the first work 1) enabling dense language models with trillions of parameters thanks to our general annotation API that supports tensor sharding with more than one dimensions 2)  showing how we can partition unevenly-shaped, non-trivial ops that have spatial dimensions with complex static configurations (e.g., convolutions with static dilation and padding). We put these enhancements in the appendix (A3.4 and A2.5) since we would rather focus on sharing what’s possible to converge and what are the bottleneck with the community, than arguing for novelty claims. The enhancements proposed by our paper are distilled from long experimental cycles and analysis, as we were relaxing the capacity bottleneck, we were at the same time facing trainability challenges one after another, and although might be seen as minor, these contributions were the key to address trainability challenges, and letting us push the scaling of neural networks to the limits.

---

### Official Review · AnonReviewer4 · 2020-10-28
**Interesting results**

**Rating:** 7
**Confidence:** 3

**Review:**

## Summary
This paper addresses the training efficiency issue of large-scale models.
The authors proposed to use GShard API to implement Transformer model with Sparsely-Gated Mixture-of-Experts, allowing sublinear scaling of the computation cost.
Experimental results on a multilingual machine translation task show that the proposed method can use computational resources efficiently.

## Strong points
* The training efficiency of giant models is one of the significant issues in the field and will attract practitioners' attention.
* The proposed model architecture is reasonable to increase efficiency. The empirical results prove their favorable scalability/efficiency trade-off.
* The results in Section 3.2, such as "scaling depth brings quality gains only after the capacity bottleneck is resolved.", suggest several interesting research directions.

## Weak points
* The readers may confuse because, although the title and abstract of the paper focus GShard, the details of GShard describe in the appendix.
* Notations are not clearly defined, so that the detail of the proposed algorithm is difficult to follow.
* Experiments using more than one trillion model weights are not ready and not shown in the paper.

## Decision reason
The empirical results are suggestive and valuable to discuss in the community.

## Questions
* What M and H stand for in the equation above Algorithm 2?
* What is the number of groups, G, and expert capacity, C, used in the experiments?  If they can be computed from other constant values, please provide the formula.

## Additional Feedback
* The operation to sets in Algorithm 1 is not clear.
  * Do Line (3) means m_e = \frac{1}{S} \sum_{s=1}^{S} g_{s, e} for all e in E ?
  * Is m_e in Line (13) defined as I described above?
  * Although other similar operations, such as c_E <- 0, are easy to guess, I recommend to define the meaning of these set operations for clarity.
* Before explaining its efficient implementation on page 3, it would be nice to explain why a sequential implementation of the gating function is required.   I guess the reason is that the assignment of an expert depends on the assignment of other experts.
* It is better to have a reference to A3.3 in the main body.  Without the reference, readers may wonder why the communication cost is the square root of the number of devices.
* Typo:
  * page 3: "our software Stack" --> "our software stack"
  * \sum_{s=1}^{s} in Line (3) of Algorithm 1 should be \sum_{s=1}^{S}

---

> ### Author Response · Authors · 2020-11-24
> **Thank you.**
>
> Thank you very much for the reviews and constructive comments.
>
> To address reviewer’s comments and feedback,
>
> 1) We updated the abstract to focus more on conditional computation.
>
> 2) We followed reviewer’s advice and dropped the set operations. We also revised the draft to define the notations more clearly. Due to space constraints, we didn’t include a table of symbols for better paper readability.
>
> 3) We did open source our GShard implementation and provided an example implementation for the GPT-3 style dense transformer. It can efficiently train transformers with up to 1T parameters and sequence length 1024.  We included a table in Appendix A3.4 to show the throughput benchmarks.  We will include a link to our open source implementation at the camera ready.
>
> 4) We dropped all the set operations/notations and replaced them with element-wise ones. Yes, Line (3) and (13) in algorithm 1 means for all the experts. We updated Algorithm 1 to clarify that.
>
> 5) Load balancing required the token assignments of one expert dependent on assignments of the other experts. Therefore, the original gating function proposed by  (Shazeer et al., 2017) had to be implemented sequentially, especially under the static shape constraints on TPUs. We revised Section 2.1 to highlight the difference between our gating function and the one proposed by (Shazeer et al., 2017). Our new gating function allows parallel execution and enables O(D) speedup with D parallel devices.
>
> 6) We added reference to A3.3 when describing the communication cost of the gating function.
>
> 7) Thanks for pointing out the typos. Fixed them in the draft.
>
> To answer the reviewer’s questions:
> 1) M stands for the model dimension and H stands for the hidden dimension in the feedforward layer, For example, the shape of wi weights is [M, H] while the shape of wo weights is [H, M]. We modified the draft to make it more clear.
>
> 2) G is proportional to D, the number of devices. In our 600B experiment, G = 2 * D = 2 *E = 4096 on 2048 accelerators. Per group expert capacity C = 16 for most models. We included these detailed numbers in appendix A.5.

---

### Official Review · AnonReviewer3 · 2020-10-29
**Great work with breakthroughs on techniques, systems and models**

**Rating:** 9
**Confidence:** 4

**Review:**

The paper develops techniques, systems and models to leverage conditional computation, greatly scaling model size while only requiring sublinear computation with respect to model size.   As a result, a sparsely-gated MoE model with 600B parameters for NMT tasks has been trained efficiently (4 days on 2048TPUs) with new state-of-art accuracy.

Merits of the paper:
- Provide an effective strategy to address the expensive computational cost of training giant models
- Offer a comprehensive solution covering techniques, systems and models
- Well-designed APIs and implementations to express a wide range of parallel computation patterns with  minimal changes to the existing mode code
-  New state-of-art models have been trained using the strategy and systems, as a great demonstration on the effectiveness and efficiency of the approach.

Places to improve
-  It is helpful to further clarify the difference of this work comparing with the prior work Shazeer et al. (2017) beside transformer vs RNN.  For example, are the features like load balancing, efficiency at scale, auxiliary loss, random routing new contributions of this work, or similar as the prior work, or with certain incremental improvements?  It would also be helpful for readers to understand, which techniques are generic and which are model-type specific.

- Both this work and the prior work Shazeer et al. (2017) apply conditional computation on neural machine translation tasks.  It would be helpful to comment on the generality and effectiveness of the solution on other types of tasks.

---

> ### Author Response · Authors · 2020-11-24
> **Appreciate the feedback!**
>
> Thank you very much for your feedback. We followed your improvement advice and revised section 2.1 in our draft to highlight our architecture innovations.
>
> 1) Our new load balancing function with fixed capacity and local dispatching improves the efficiency of gating function by O(D) times for D parallel devices.
>
> 2) The auxiliary loss we introduced is differentiable, which provides better numerical stability since it can now be optimized with gradient descent.
>
> 3) For different second expert policies, we did a brief study with smaller scale models, we observed less overflowed tokens with random 2nd expert selection.  We couldn’t do ablation study since training is costly so we picked the tradeoff at small scale and used it for all other scales.
>
> Those innovations can be applied to MoE gating functions with arbitrary expert architectures.
>
> We also discussed the application of the conditional computation on other tasks such as language modeling and computer vision in the added related work section.

---

### Decision · Program_Chairs · 2021-01-07
**Final Decision**

**Decision:**

Accept (Poster)

**Comment:**

This paper is a study of neural network scaling, with models containing hundred of billions of parameters. To that end, the paper introduce a new module called GShard, consisting of annotations APIs on how to split computations across accelerators, which is integrated in the XLA compiler. This enables the training of models with hundreds billions of parameters. To scale efficiently to very large models, the paper proposes to use transformer networks, where every other feed forward sub-layer is replaced by a sparse mixture of experts (similar to Shazeer et al. 2017). This model is then evaluated on a multilingual machine translation task, from 100 languages to English.

On the one hand, I believe that the contributions of the paper are significant: scaling to 600B parameters, and showing that this leads to better translation quality are important achievement. The analysis of transformer networks scaling could also have an important impact. Finally I think that GShard and its integration in XLA could be very valuable. On the other hand, I agree with some of the concerns raised by the reviewers, regarding the writing of the paper and the reproducibility. I found the paper not well written, and hard to identify the differences with previous work. As GShard is one of the main contribution, I would expect a better description of it in the main text (compared to the MoE which seems more incremental). Regarding reproducibility, I do not think that the authors provided a good reason not to evaluate on standard benchmarks: the test sets could be excluded from the train set through various deduplication heuristics.

To conclude, I am leaning toward accepting the paper, but believe it is borderline. The reason is that the contributions are significant, and worth publishing. But I would not oppose a rejection based on the reproducibility and writing issues.